# Graphene-Based Inks for Printing of Planar Micro-Supercapacitors: A Review

**DOI:** 10.3390/ma12060978

**Published:** 2019-03-25

**Authors:** Tuan Sang Tran, Naba Kumar Dutta, Namita Roy Choudhury

**Affiliations:** Department of Chemical and Environmental Engineering, School of Engineering, RMIT University, Melbourne, Victoria 3000, Australia; sang.tran2@student.rmit.edu.au (T.S.T.); naba.dutta@rmit.edu.au (N.K.D.)

**Keywords:** graphene inks, micro-supercapacitors, energy storage, printing techniques, microelectronics

## Abstract

Micro-supercapacitors have recently emerged as promising microscale power sources for portable and wearable microelectronics. However, most reported planar micro-supercapacitors suffer from low energy density and the complexity of fabrication, which calls for their further development. In recent years, the fortification of graphene has enabled the dramatic improvement of planar micro-supercapacitors by taking full advantage of in-plane interdigital architecture and the unique features of graphene. The development of viable printing technologies has also provided better means for manufacturing, bringing micro-supercapacitors closer to practical applications. This review summarizes the latest advances in graphene-based planar micro-supercapacitors, with specific emphasis placed on formulation of graphene-based inks and their fabrication routes onto interdigital electrodes. Prospects and challenges in this field are also discussed towards the realization of graphene-based planar micro-supercapacitors in the world of microelectronics.

## 1. Introduction

The push towards internet of things (IoT) may become a key technological and economical driver for global development in the near future. IoT allow for sensors and actuators (devices) to deploy over a large area, connect into large databases and networks (internet), and autonomously operate in correlation with computing systems [1]. The maturation of IoT will find importance, not only in the retail market (wearable electronics), but also in manufacturing, infrastructure/environment monitoring, healthcare, transportation, and so forth [1,2,3]. Monitoring of environmental pollutants, for example, can be a tremendous task due to the scale and mobility of the environment. A network of thousands of microsensors, which deploys over a large area and autonomously collect the data, can effectively detect any environmental issues and provide the precise information in real time. A critical requirement for these systems is power autonomy for independent and maintenance-free operation. Currently, the available microscale power sources are mainly relying on microbatteries, which possess low power density and limited lifetime [4]. Due to their low power density, a series of combined microbatteries is usually required to provide sufficient energy to power the microsystem, making them much larger than the device they are to power [5]. Replacing batteries at the end of their life is also a major problem. Therefore, the search for alternative and sustainable microscale energy storage devices has attracted enormous attention. 

Unlike batteries, which store energy and produce electricity through chemical reactions [6,7], supercapacitors store electrical energy directly via reversible adsorption–desorption of ions at the electrode/electrolyte interface [8] or pseudo-capacititive redox reactions between electrodes and electrolyte [9]. As a result, supercapacitors have a number of advantages including high power density, fast charge/discharge rate, and long cycle life [10,11]. With these excellent properties, supercapacitors hold much promise as an efficient alternative for batteries in various applications. As conventional supercapacitors are too large to be adapted into microelectronics, research have driven towards the design and fabrication of miniaturized supercapacitors (micro-supercapacitors) as an effort to replace microbatteries for powering microelectronics. The flexibility and performance of micro-supercapacitors can be tuned by choosing suitable materials and designs [12,13]. Compared with the traditional sandwich structure, micro-supercapacitors usually choose a planar design with in-plane interdigital electrode finger arrays due to their numerous advantages in performance and fabrication [14]. In this review, we mainly discuss the advances in micro-supercapacitors with planar architecture as they are mostly suitable for powering portable and wearable microelectronics. 

Since the discovery of graphene a decade ago, a great numbers of its potential uses have been proposed [15,16,17]. Owing to its extraordinary high surface area of up to 2630 m^2^·g^−1^ and exceptional carrier mobility of up to 2 × 10^5^ cm^2^·V^−1^·cm^−2^ [18,19], graphene is one of the most promising materials to store electrical charge to date [20,21]. In recent years, graphene has shown great potential in energy storage devices, particularly in micro-supercapacitors [21,22]. The quantum capacitance of graphene was reported to be ~21 µF·cm^−2^ (~550 F·g^−1^) [23], which is among the best electrode materials for supercapacitors [10,24]. The macroscopic structure of graphene can be manipulated from its original 2D sheets into new graphene architectures to enhance its electrochemical properties (Figure 1). Various graphene nanostructures, such as wrinkled graphene [25], porous graphene [26], graphene nanomeshes [27], honeycomb-like graphene [28], graphene hydrogels [29], and 3D porous graphene [30], have been reported with improved electrochemical performance. In recent years, significant interest has been devoted to the assembly of vertically oriented graphene [31,32], which not only possesses exceptionally high surface area, but also provides accessible paths for the fast adsorption and desorption of ions, leading to micro-supercapacitors with very high energy densities and ultrafast response times [33,34]. Graphene is not only advantageous by itself, but also promising for combining with other materials to boost their superior performance [35,36]. The use of graphene has opened up many new features for micro-supercapacitor devices that did not exist before, such as ultrathin, flexible, rollable, transparent, and beyond. 

To realize the commercial application of graphene-based micro-supercapacitors, it is necessary to develop a facile, reliable, and cost-effective technique for scalable fabrication of graphene electrodes. Among the available processing techniques, printing of graphene inks offers a simple and effective route for production of interdigital electrodes that can be adapted into an industrially accessible scale [37]. In this review, we aim to guide the readers through recent advances in graphene inks for printing of planar micro-supercapacitors. First, we briefly discuss the fundamentals of micro-supercapacitors, including materials, designs, and performance evaluation. Then, we review the current formulation of graphene inks in correlation with the printing technologies for fabrication of in-plane interdigital electrodes. Finally, we give an insight into the challenges and outlook of graphene inks for the future development of micro-supercapacitors.

## 2. Fundamentals of Micro-Supercapacitors

For supercapacitor devices, the capacitance (C), energy density (E), and power density (P) can be calculated according to the following formulae:(1)$$C=\frac{i}{\left(-dV/dt\right)}$$
(2)$$E=\frac{1}{2}CV^{2}$$
(3)$$P=\frac{E}{t}$$
where i is the applied current, dV/dt is the slope of the galvanostatic discharge curve (CC curve), V is the operating voltage window, and t is the discharge time [38,39]. According to these equations, the performance of supercapacitor devices can be theoretically improved by increasing the capacitance, broadening the operating voltage, and reducing the discharge time. These parameters provide fundamental guidance for choosing materials and designs of micro-supercapacitors.

### 2.1. Materials

The basic structure of micro-supercapacitors consists of four main components—a substrate, current collectors, electrodes, and electrolytes (in many cases, this includes a separator). The energetic performance of micro-supercapacitors is mainly dependent on the intrinsic electrochemical properties of the electrode materials [24]. Based on the charge-storage mechanisms, electrode materials can be classified into two categories—electric double-layer capacitive (EDLC) materials and pseudo-capacitive materials. For EDLC materials, charges are electrostatically stored at the interface between electrode and electrolyte [8]. Therefore, materials with high specific surface area and good electrical conductivity, such as activated carbon [40], onion-like carbon [41], carbon nanotubes [42], and graphene [43,44], are preferable for active electrode materials. On the other side, metal oxides (such as Co_3_O_4_, MoO_3_, MnO_2_, NiO, and RuO_2_) [45,46,47] and conductive polymers (such as polythiophene, polyaniline, and polypyrrole) [48,49] can provide a much higher intrinsic capacitance via reversible pseudo-capacitive redox reactions between electrodes and electrolyte, leading to higher energy density [9]. Generally, EDLC materials can offer fast charge/discharge rate with a long life-cycle of up to millions of times but have relatively low energy density. Meanwhile, pseudo-capacitive materials can deliver more energy but have a slow rate and limited lifetime. Both types of these materials, and their composites, have been widely used as active electrode materials and gained significant achievements in the development of micro-supercapacitors [49,50,51].

Another key factor affecting the performance of micro-supercapacitors is the electrolytes, which provide ions for the charge-storage mechanisms and define the operating voltage window of the devices [52,53]. Liquid electrolytes (aqueous, organic, or ionic) are widely used in supercapacitors with conventional sandwich structure due to their high ionic conductivity [54]. However, liquid electrolytes are not suitable for planar micro-supercapacitors because they are difficult to encapsulate and suffer from leakage problem. Therefore, solid-state electrolytes have emerged as feasible alternatives for the liquid derivatives [55]. Aqueous-based solid-state electrolytes can only be operated in a potential window of less than 1 V due to water electrolysis [43], while ionic liquid-derived solid-state electrolytes can operate in potential window of up to 2.5 V [56,57], providing higher energy density [58,59]. Solid-state electrolytes can, not only solve the leakage problem, but also offer greater reliability, wider range of operating temperature, and extra features such as flexibility and stretchability for micro-supercapacitor devices. 

Other components, such as substrates and current collectors, may also affect the flexibility and reliability of micro-supercapacitors, but the intrinsic capacitance and the amount of power it can deliver are among the most important facets when considering the performance of micro-supercapacitors.

### 2.2. Designs

The basic design of micro-supercapacitors can be divided into two categories: sandwich and planar configuration (Figure 2). The structure of early-stage micro-supercapacitors was inspired from thin-film microbatteries [60,61], where two thin-film electrodes are deposited on the current collectors and sandwiched between the electrolyte (Figure 2a). The first demonstration of micro-supercapacitors was reported by Lim and co-workers [61] in 2001, in which two ruthenium oxide (RuO_2_) thin films were sandwiched between lithium phosphorousoxynitride (LiPON) solid electrolytes, and exhibited a volumetric capacitance of ~380 µF·cm^−3^. This conventional sandwich structure is preferable for cost-effective mass production, since it inherited the fabrication technologies from thin-film microbatteries [12]. However, from the practical applications viewpoint, it suffers from some significant drawbacks such as limited flexibility, possibility of short circuit, and undesirable position dislocation of electrodes [13]. It is also challenging to accurately control the thickness of the separator and electrolyte, which may increase ion transport resistance and lead to the degradation of power [54]. On the other side, planar micro-supercapacitors with in-plane interdigital finger arrays design have more advantages in flexibility, reliability, and fabrication (Figure 2b). The planar design was implemented early by Sung et al. [62] in 2003 by filling the gap (~50 μm) between polypyrrole (PPy) and poly-(3-phenylthiophene) (PPT) electrode arrays with liquid electrolytes, which resulted in supercapacitor cells with capacitance of ~5.2 mF. Planar micro-supercapacitors can be constructed using numerous fabrication techniques, from conventional printing to advanced micropatterning [13,37], which help to drive the cost down for commercialization. The side-by-side electrode finger designs allow for more flexible and reliable devices without the fear of short-circuit or electrode dislocation under various application conditions. With the development of the micro-fabrication techniques, the interspace between electrodes can be narrowed down to several hundred nanometers [63], smaller than the thickness of the separator and electrolyte layer in the sandwich design [64,65]. As a result, the ion diffusion paths can be effectively shortened, which reduces the charge/discharge time and leads to higher power capacity. The thickness of planar micro-supercapacitors can be slenderized by choosing thinner electrode materials and substrates [66], or can even be engineered down to negligible by patterning directly on to the surface of the devices [67]. All these merits make planar micro-supercapacitors promising candidates for on-chip integration and powering microelectronics.

### 2.3. Performance Evaluation

For benchmarking of micro-supercapacitors, the traditional yardstick to evaluate their performance is calculating their capacitance, energy, and power densities based on the weight and/or volume of the devices. However, unlike conventional supercapacitors, the mass of the electrode materials in micro-supercapacitors is almost negligible compared to the weight of the devices. As the active electrode layers are too thin (micro/nanometer scales) and their thicknesses may not be uniform throughout the devices, volumetric measurements may provide misleading information about their performance. Hence, neither gravimetric nor volumetric properties should be used to evaluate the performance of micro-supercapacitors, especially for those with planar architecture. In contrast to the weight and volume, the footprint area of the devices is actually the key concern for micro-supercapacitors. Therefore, the proper way for reporting micro-supercapacitors performance is to normalize its features (capacitance, energy, and power densities) by the footprint area of the devices (per cm^2^). Kyeremateng and colleagues [68] have proposed a standardized metric for reporting the performance of micro-supercapacitors. In sandwich configuration, the device consists of two stacked electrodes, but the footprint area is confined by the surface of only one electrode. Therefore, the areal capacitance of the cell $C_{sandwich}$ is only half of the individual capacitance of each single electrode ($C_{sandwich}=C/2$). In planar configuration, the footprint area includes the surface area of both electrodes and the inactive gap between them. Even if the inactive gap is minimized, the surface of each electrode will be less than half of the confined area. Therefore, the cell capacitance is less than one-fourth of the areal capacitance of the individual electrode ($C_{planar}<C/4$). Apart from energetic performances, other characteristics such as the cyclability, flexibility, charge/discharge rate, operating potential, and operating temperature are also important facets when comparing planar micro-supercapacitors.

## 3. Graphene-Based Inks for Electrode Materials

Graphene can be obtained from a plethora of methods, which can be classified as either “bottom-up” or “top-down” strategies [69]. The bottom-up approach is based on the epitaxial growth of two-dimensional carbon layers by chemical vapor deposition, which is costly and unable to upscale for industrial production [70]. Meanwhile, the top-down methods, including exfoliation of graphite and reduction of graphene oxide, are widely used for the production of graphene due to its cost-effectiveness and solution processability [71].

Solution processing offers a facile route for production of graphene and can be further adopted by the current printing techniques used in the industry for the fabrication of interdigital electrodes [71,72]. There are three main requirements for printable graphene-based dispersions: (i) Homogeneous and stable against precipitation, (ii) compatible fluidic properties (viscosity and surface tension) with the printing devices, and (iii) ecofriendly and low boiling-point solvent for ease of processing. In the following section, we will discuss the current formulation of graphene inks that are feasible for printing.

### 3.1. Pristine Graphene

Beside the bottom-up methods, which can produce graphene with “pristine” quality but limited quantity, liquid-phase exfoliation of graphite is considered as the most effective route for large scale production of pristine graphene. The principle of liquid-phase exfoliation is based on overcoming the van der Waals attractions between stacked adjacent graphene layers by liquid immersion under sonication force or high shear rate [73,74]. According to the dispersive London interactions [75], the potential energy between adjacent layers is significantly reduced when immersed in a liquid medium, leading to the idea of using solvents to extract graphene from its stacked form (graphite). A recent study by Coleman and co-workers [76] indicated that graphene can be effectively exfoliated from graphite using solvents such as N-methyl-2-pyrrolidone (NMP) and N,N-dimethylformamide (DMF), setting up the background for formulation of solvent-based pristine graphene inks.

In 2013, Li and co-workers [77] formulated high-concentration and stable graphene inks by ultrasonication of graphite in DMF, which is compatible for inkjet printing. The inks were printed and annealed at 400 °C for few hours, achieving graphene patterns with excellent electrical conductivity. More recently, Majee and colleagues [78] employed a L5M Silverson mixer for shear exfoliation of graphite in NMP, formulating a highly concentrated and stable graphene ink (3.2 mg/mL). The graphene ink composed of 4-layer graphene flakes with uniformly lateral size of ~160 nm was then inkjet-printed on a glass substrate and annealed at ~350 °C for 150 min, which resulted in near-transparent and conductive graphene circuits, which is applicable for printing of interdigital electrodes for micro-supercapacitors.

However, the use of these solvents poses significant issues including the high cost, the high annealing temperature, and the toxicity to both human and the environment [79,80]. Therefore, research has been driven towards low boiling-point and environmentally benign solvents. Common solvents such as acetone, ethanol, and isopropanol usually come up with unsuitable surface energy, leading to poor graphene dispersions [37]. Hence, stabilizers are usually added to support the exfoliation of graphene in these liquid mediums.

By using ethyl cellulose as a stabilizer, Secor et al. [81] developed a novel graphene ink by liquid-phase exfoliation of graphite in ethanol, an environmentally benign solvent. The ink has a viscosity of ~0.01 Pa s and a surface tension of ∼33 mN/m, compatible to inkjet printing. Gao and colleagues [82] also reported a new route for formulation of ethyl cellulose-stabilized pristine graphene ink by direct exfoliation from graphite using ultrasound-assisted supercritical CO_2_ (Figure 3). The ink was printed using inkjet printing and resulted in graphene patterns with extremely high conductivity, which is promising for fabrication of interdigital electrodes in planar micro-supercapacitors.

In 2016, Arapov and co-workers [83] described an approach for the preparation of highly concentrated graphene inks for screen-printing (Figure 4). The ink pastes were prepared by high-shear mixing of expanded graphite in the presence of a polymeric binder, followed by mild heating to trigger gelation of graphene/polymer dispersions, which resulted in colloidally stable and highly concentrated graphene pastes (52 mg·mL^−1^) that showed excellent performance in screen printing. The printed patterns with line resolutions of ~40 µm were dried at 100 °C for only 5 min and exhibited excellent sheet resistances of 30 Ω/sq at 25 μm thickness. Hyun et al. [84] also formulated highly viscous graphene inks by dispersing graphene with ethylcellulose in ethanol and terpineol. The resulted inks showed shear viscosities of 1–10 cP and good performance in screen printing. These formulations have not been used for screen printing of micro-supercapacitors, and the electrochemical performance of the printed patterns still remains unknown. However, they have enormous potential in high-volume roll-to-roll fabrication of interdigital electrodes for planar micro-supercapacitors.

Water has long been established as the most preferred solvent due to its low boiling point and non-toxic nature. As graphene cannot be dispersed in water alone due to its hydrophobicity, surfactants are usually used to tailor their interfacial energy and stabilize the exfoliated graphene flakes against aggregation [85,86,87]. By introducing sodium cholate into water, Lotya et al. [85] have successfully produced aqueous graphene dispersions with high concentrations of up to 0.3 mg/mL. The prepared dispersions are highly stable and can be easily casted into various substrates, making them prospective for printing of interdigital electrodes for planar micro-supercapacitors. A range of ionic [85,88], non-ionic [86,88], polymeric [89], and bio-surfactants [90] were reported to be effective for preparation of aqueous graphene dispersions, which are ideal for formulation of pristine graphene inks [37]. Further research should focus on construction of various graphene architectures in the printed electrodes and investigation of their electrochemical performance.

### 3.2. Graphene Oxide

Graphene oxide (GO) is produced by the oxidative treatment of graphite via either Brodie [91], Staudenmaier [92], Hummers [93], or some variation of these methods [94]. It contains a range of oxygen-functional groups, which trigger its hydrophilicity and solution processability, and can be reduced to form graphene-like materials [18]. In fact, the majority of studies on graphene and its application are not based on pristine graphene, but rather the reduced graphene oxide (rGO) [37]. This is because of the ease of production and the capability to render its functionalities. 

The formulation of GO inks is simple and straightforward as GO can be easily dispersed in the most preferable solvent, water. In 2011, Le et al. [95] demonstrated that the dispersions of GO in water is stable and compatible to inkjet printing. The GO inks were inkjet-printed and thermally reduced at 200 °C for 12 h under N_2_ atmosphere, forming conductive graphene electrodes with the spatial resolution of ~50 μm. Likewise, Shin and co-workers used an inkjet printer for micropatterning of aqueous GO inks with different concentrations onto poly(ethylene terephthalate) (PET) substrates. The printed GO patterns were reduced in a chamber containing hydrazine and ammonia solution at 90 °C for 1 h, which resulted in conductive graphene electrodes with excellent conductivity of ~65 Ω/sq. These works paved a new avenue for the fabrication of graphene electrodes for micro-supercapacitors.

For 3D printing, the formulation of graphene inks is relatively different, as a printable ink solution required a high viscosity and shear-thinning behavior. In 2017, Rocha and co-workers [96] formulated printable GO inks in aqueous Pluronic F127, a thermoresponsive polymer, for 3D printing. As F127 formed hydrogels in water [97], a stable and concentrated graphene colloidal system could be achieved. The formulated ink had a high viscosity and exhibited shear-thinning behavior, which was printed through a micronozzle for fabrication of supercapacitor electrodes. The printed electrodes were lyophilized for 48 h and thermally reduced at 900 °C for 1 h under H_2_/Ar atmosphere, which showed good electrochemical performance and achieved a capacitance of up to 140 F·g^−1^.

Interestingly, among all studies on the chemistry of GO, the largest portion is focused on its reduction routes back to graphene [35,98]. As GO is not electrically conductive, it need to be reduced to graphene by either thermal [99], chemical [100], or photothermal [101] routes to restore its electrical conductivity before it can be used as electrodes for supercapacitor. Different reduction methods and their performance in graphene-based planar micro-supercapacitors are summarized in Table 1.

By nature, hydrated graphene oxide is simultaneously an electrical insulator and a good ionic conductor, allowing it to serve as electrolyte and separator in energy devices. Hence, it is possible to produce all-graphene micro-supercapacitors by employing rGO as electrodes and GO as electrolyte. In 2011, Gao and co-workers [106] demonstrated the ability to fabricate graphene micro-supercapacitors on hydrated GO films using laser irradiation, which can work without the use of external electrolytes (Figure 5). In this work, free-standing GO films were made by vacuum filtration. By selective reduction of GO films using a CO_2_ laser, conductive rGO arrays with porous structure were formed and served as active electrodes, while the intact GO served as electrolyte. Micro-supercapacitor devices with both sandwich and in-plane architecture were fabricated and showed comparable performance with those using external electrolytes. This work is significant since it removed the need of additional electrolyte and established a new approach for scalable fabrication of all-carbon, high-precision, and lightweight micro-supercapacitors.

### 3.3. Graphene Composites

Graphene is not only advantageous by itself but is also promising for combining with other materials to boost their superior performance. Intensive research has been devoted to the design and synthesis of graphene hybrid complexes for enhancing their electrochemical properties [107,108]. By mixing graphene with carbon nanotubes [109], conductive polymer [110], or transition metal oxides [51], the composite inks can take full advantage of the two materials to enhance their electrochemical performance.

One of the most apparent challenges when processing graphene inks is the restacking of graphene layers, which leads to lower active surface area and the degradation of power capacity. Yang et al. [109] have successfully inhibited stacking of individual graphene sheets by introducing one-dimensional carbon nanotubes (CNTs) into graphene dispersions to form 3D hierarchical porous structure. The presence of CNTs as nanospacers effectively enlarged the space between graphene sheets, increased the active area for charge storage, and enhanced the energetic performance of the supercapacitor device. The ink was deposited onto a graphite substrate (1 cm^2^) to form the test electrodes. A supercapacitor device was fabricated and exhibited specific capacitance of 326.5 F·g^−1^ with ultrahigh energy and power densities (21.7 Wh·kg^−1^ and 78.3 kW·kg^−1^, respectively). 

Poly (3,4-ethylenedioxythiophene): poly (styrenesulfonic acid) (PEDOT:PSS) is an important conductive polythiophene derivative and a favorable electrode material for supercapacitors. Liu and colleagues [111] have successfully formulated graphene/PEDOT:PSS hybrid inks for direct printing of high-performance micro-supercapacitors (Figure 6). The presence of PEDOT:PSS not only stabilized graphene, but also enhanced its electrochemical properties. The printed micro-supercapacitor on a paper substrate exhibited a superior areal capacitance of 5.4 mF·cm^−2^, which is among the highest value achieved on graphene-based micro-supercapacitors.

Polyaniline (PANi), a typical conducting polymer with pseudocapacitance, is commonly used as electrode material for supercapacitors. In 2014, Xu et al. [110] formulated graphene/PANi inks for inkjet printing of supercapacitor electrodes. The composite inks were prepared by SDBS surfactant assisted sonication of graphite powder and polyaniline in water. The composite inks were inkjet-printed and annealed at 80 °C for 2 h and exhibited excellent conductivity of 0.29 S·cm^−1^. A supercapacitor cell was fabricated using the printed electrodes and yielded a maximum specific capacitance of 82 F·g^−1^, power density of 124 kW·kg^−1^, and energy density of 2.4 Wh·kg^−1^. By growing vertically aligned pseudo-capacitive PANi nanorods on both sides of the GO surface and subsequent reduction in the presence of PEDOT:PSS, Liu and co-workers [112] formulated highly concentrated, highly viscous, and water-dispersible composite inks for extrusion printing (Figure 7). The printed all-solid-state micro-supercapacitors exhibited outstanding areal capacitance of 153.6 mF·cm^−2^ and volumetric capacitance of 19.2 F·cm^−3^. By adapting an asymmetric design, the printed micro-supercapacitor with extended operating voltage window from 0.8 to 1.2 V achieved an improved energy density (from 3.36 to 4.83 mWh·cm^−3^) and power density (from 9.82 to 25.3 W·cm^−3^).

There is still a scarcity of literature of graphene composite inks for printing of micro-supercapacitors. In fact, numerous pseudocapacitive materials have been reported to have higher capacitance when made of composites with graphene, including conductive polymers (such as polyaniline, polypyrrole, and polythiophene) [48,49] and metal oxides (such as MnO_2_, MoO_3_, Co_3_O_4_, NiO, and RuO_2_) [45,46,47]. However, most of these graphene complexes have only been realized by other micro-fabrication techniques, not printing. Therefore, it still requires a great deal of effort to pioneer graphene composites into printable ink dispersions for scalable fabrication of high-performance graphene-based micro-supercapacitors.

## 4. Printing Techniques

Printing technologies have been widely employed for fabrication of microelectronics [113]. Compared with other microfabrication techniques that involve complicated processes and harsh operation conditions [114], printing of graphene inks offers feasible routes for fabrication of interdigital electrodes onto a myriad of substrates with low cost and high versatility. Until now, a number of mass printing techniques have been developed for processing of graphene dispersions into electrodes such as screen printing [65,115], inkjet printing [116,117], and 3D printing [118,119]. Several important aspects of these techniques are summarized in Table 2 for better comparison.

### 4.1. Screen Printing

The screen-printing process is based on the penetration of ink pastes through the patterned mask/stencil under pressing force of a squeegee [84]. Among the available printing techniques, screen printing is considered as the most facile and cost-effective route for mass printing of planar micro-supercapacitors [37,120]. In 2014, Liu and co-workers [121] used screen printing to fabricate flexible all-solid-state micro-supercapacitors using N-doped reduced graphene oxide (rGO) as the electrode material. Important physicochemical properties of the inks, such as viscosity, surface tension, or shear-thinning behavior, were not reported. The formulated inks were successfully screen-printed into interdigital electrodes with the active area of 0.396 cm^2^ and the thickness of 10 μm, which were further coated with a layer of PVA-H_3_PO_4_ as solid-state electrolyte. The printed micro-supercapacitor delivered a high specific areal capacitance of 3.4 mF·cm^−2^ with good rate capability and cycling stability (Figure 8). More recently, Shi and colleagues [122] have developed an industrially applicable screen-printing protocol for low-cost production of ultrahigh-voltage integrated micro-supercapacitors with designable shapes, aesthetic versatility, outstanding flexibility, and superior modularization. The inks for screen printing were prepared by mixing high-quality graphene, conducting carbon black, and poly(vinyl chloride-co-vinyl acetate) binder (P-VC/VAc) in dimethyl mixed dibasic acid ester (DBE) solvent. This formulation resulted in graphene inks with outstanding shear-thinning behavior, allowing for extrusion of the ink through the stencil and quick solidification without shear force, with ideal physicochemical properties for screen printing. A tandem pack of 130 micro-supercapacitor cells combined was fabricated and delivered a remarkable voltage of more than 100 V, demonstrating the robustness of the protocol and the printing technique.

The quality of the screen-printed patterns is mainly defined by the quality of ink pastes and the resolution of the stencil [123]. An ideal ink paste should have high rest viscosity and shear-thinning behavior [124,125]. The viscosity of a screen-printable ink can be varied from 0.05 to 5 Pa·s [72]. Therefore, it is required to formulate highly concentrated graphene dispersions to meet the viscosity requirements. Owing to its simple operating principle, screen printing is faster in comparison to other printing tools, making it an eminent candidate for mass production of interdigital electrodes with low cost and high throughput [126].

Screen printing also has several drawbacks. As the ink pastes for screen printing are highly concentrated, it could be dried out during the printing process, negatively affecting the stencil and the desired patterns. The direct contact between stencil and substrates also prevents the ability for micro-supercapacitors to be printed directly onto the surface of microelectronics. It is also challenging to produce stable and concentrated graphene dispersions without aggregation. To meet the rheology requirements for a printable ink, future formulations should focus on either preparation of highly concentrated graphene emulsions or gelation of graphene in polymeric matrixes. It is no doubt that screen printing will be a strong candidate for high-volume roll-to-roll production of micro-supercapacitors.

### 4.2. Inkjet Printing

Unlike screen printing, inkjet printing works without the need of a physical printing mask [127]. The basic principle of inkjet printing is the ejection of micro-sized ink droplets through a micro-nozzle onto their accurate position on the substrate to form the desired patterns [128]. The micro-droplets can be generated by either thermal or piezoelectric excitation [129]. For accurate positioning, the ink droplets can be driven by electrostatic force (continuous mode) or selectively triggered whenever the nozzle reach its appropriate position (drops-on-demand mode) [37,130] (Figure 9). In 2017, Li et al. [131] developed a simple full-inkjet-printing technique for scalable fabrication of graphene-based micro-supercapacitors. By solvent exchange technique, high-concentration electrochemically exfoliated graphene (EEG) inks were formulated and efficiently used for inkjet printing. Thick graphene patterns (with thickness up to ∼0.7 μm) were successfully printed in a scalable and large manner, which could serve as both the electrodes and current collectors. An electrolyte formulation of poly(4-styrenesulfonic acid), phosphoric acid, and ethylene glycol with suitable rheology was then printed onto the as-printed graphene electrodes and gelled, which resulted in fully printed solid-state graphene-based micro-supercapacitors (areal capacitance of 0.7 mF·cm^−2^). This technique is significant since it removed unnecessary steps, provided a promising route for scalable and fully automated fabrication of micro-supercapacitors.

Inkjet printing of graphene inks has attracted enormous attention due to its high resolution and versatility. The most simple and straightforward formulation of graphene inks is based on graphene oxide (GO inks) due to its hydrophilic nature to form stable dispersion in water and solvents, as in a report by Le et al. [95] in 2011. Several kinds of pristine graphene inks [78,81,82] and graphene hybrid inks [110,132,133] have also been successfully integrated with inkjet printing and showed great performance. The resolution of inkjet printing can reach ~2 µm without great difficulty [116]. The non-contact manner also opens up the opportunity to print micro-supercapacitors directly onto the surface of the microdevices. 

In recent years, research efforts have been emphasized on the modification of the fluidic characteristic of graphene inks, since they are crucial factors for printable ink systems. Generally, ink dispersion with low viscosity (0.004–0.03 Pa·s) and high surface tension (typically ~35 mN·m^−1^) are required for the formation of ink droplets [130]. For the formulation of graphene inks, the lateral size of graphene flakes and its stability in the dispersions are also the main concern as they can block the pin hole of the nozzle from jetting droplets [127]. Inkjet printing of different types of graphene inks should also take the homogeneity of the dispersions into consideration. This technique is ideal for fabrication of ultrathin planar micro-supercapacitors since it allows for the deposition of a very thin graphene patterns. As demonstrated by Secor et al. [81] in 2013, ultrathin graphene patterns can be achieved with the thickness of less than 50 nm by a single printing pass. However, when fabricating thicker devices, multiple printing passes may be required, decreasing the process throughput.

### 4.3. 3D Printing

3D printing is a new approach for advanced manufacturing in which materials are deposited layer-by-layer to produce three dimensional objects [134]. The process of 3D printing usually involves the extrusion of inks/filaments through a micro-nozzle, which is computationally controlled by a three-axis motion stage [135]. When solidified, it forms three-dimensional objects with controlled geometry and porosity (Figure 10). In recent years, 3D printing has emerged as a viable route for fabrication of energy devices, particularly for the fabrication of planar micro-supercapacitors [136,137]. Among 3D-printing techniques, fused depositing modeling (FDM) is one the most commonly used technique since it allows for the use of various kinds of materials. Zhang and colleagues [138] used melt blending to homogeneously disperse rGO into polylactic acid (PLA), which was processed into 1.75 mm diameter filament. By melt extrusion of the rGO/PLA filament, highly conductive and flexible circuits were 3D-printed onto different substrates, well suited for fabrication of micro-supercapacitors. In recent years, extrusion-based 3D printing, where viscous inks are selectively extruded through a nozzle, has attracted enormous attention. In 2017, Rocha and co-workers [96] demonstrated a 3D-printing technique based on the continuous extrusion of colloidal inks at room temperature. GO and Cu/GO composite inks were formulated in thermoresponsive polymer Pluronic F127 aqueous solutions. As F127 formed hydrogels in water [97], it could carry any particle in the system, such as graphene. The inks had high viscosity and shear-thinning behavior, which could be printed through the nozzle for fabrication of supercapacitor electrodes. More recently, Liu et al. [112] demonstrated extrusion printing of GO/PANi hybrid inks for the fabrication of high-performance micro-supercapacitors. As extrusion printing required a stable, highly viscous, and homogeneous dispersion of active materials, PEDOT:PSS was used in the formulation as a dispersing agent in order to meet the stability requirement of extrusion printing. The composite ink had shear-thinning behavior and a higher viscosity of ~6.1 Pa·s at 1 s^−1^, within the range of extrusion printing. Symmetric and asymmetric micro-supercapacitors were successfully printed and exhibited remarkable performance.

In 3D printing, the quick solidification and shear-thinning behavior of the inks are critical factors defining the quality of the printed objects [139,140]. Unfortunately, graphene dispersions in neat solvents do not exhibit these properties [140,141]. Hence, formulations of graphene inks for 3D printing usually require the implementation of viscosifiers and gelable fillers to tailor its printability [142,143,144]. However, these additives usually do not possess any interesting electronic properties, which may negatively affect the energetic performance of the printed micro-supercapacitors [142,145]. The resolution of 3D printing is mainly dependent on the size of the nozzle and the fluidic properties of the inks. The highest resolution achieved by 3D printing was ~10 µm, similar to those of screen printing [65,119]. Research on 3D printing has drawn enormous attention, and it is clear that this technique will find importance in the fabrication of planar micro-supercapacitors.

## 5. Conclusions and Outlook

In this review, we have highlighted the most recent works on the formulation of graphene inks and the printing techniques used for fabrication of planar micro-supercapacitors. Printing technologies offer promising solution for mass production of interdigital electrodes, bringing this miniaturized power source closer to practical application. 

It appears that a number of innovative approaches have been adopted for formulation of graphene inks. However, research on graphene inks are still mainly focused on formulation of stable graphene dispersions with printability and demonstration of simple conductive circuits, while less attention has been paid to their electrochemical performance. With the increasing amount of research in this field, it is crucial to uniformly report their performance for more reliable comparison of different micro-supercapacitors. As the key concern of these microdevices is their footprint area, reporting the performance of micro-supercapacitors should focus on the real metric of the devices.

Although researchers have demonstrated printed micro-supercapacitors with good performance, the cost and the environmental concerns still limit their practical realization. The production of graphene oxide is not environmentally friendly, which involves harsh oxidation and reduction processes. Most of the graphene composites for energy storage applications to date are also based on graphene oxide, and, thus, raise the concern about their sustainability. Pristine graphene inks still suffer from the high annealing temperature and the use of harsh solvents. Water-based pristine graphene inks are preferable and more sustainable, but a third added component (surfactants or stabilizers) is required for stabilizing graphene, which usually does not exhibit any interesting electrochemical properties. Therefore, the combination of pristine graphene with other materials (conducting polymers or metal oxides), which can both stabilize graphene and exhibit pseudocapacitance, can be a good direction for future development. 

Further research effort should also be devoted to remedy the printability of the devices and enhance the printing resolution. More advances still need to be made in the engineering of compatible graphene inks and the printing protocols for cost-effective fabrication, as the nanostructures of graphene in the printed electrodes play an important role in the performance of micro-supercapacitors. Further research should also be devoted to the investigation of various graphene nanostructures in the formulation of new graphene inks and their electrochemical performance. Research in this field is still at the nascent stage, but there is no doubt that the printing of graphene inks can be a promising key to open the door for graphene-based planar micro-supercapacitors in the future.

## Figures and Tables

**Figure 1 materials-12-00978-f001:**
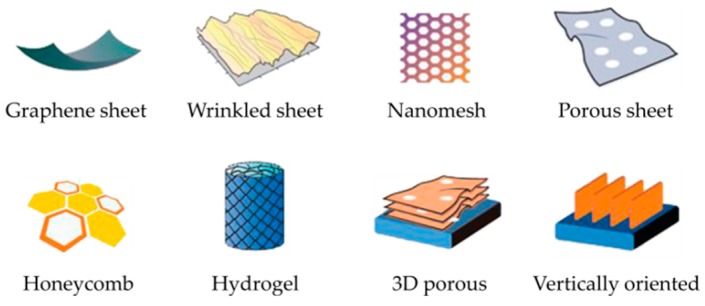
Schematic illustration of some typical graphene macroscopic structures that are useful for energy storage applications. Adapted with permission from Reference [21]. Copyright © 2016 Springer Nature.

**Figure 2 materials-12-00978-f002:**
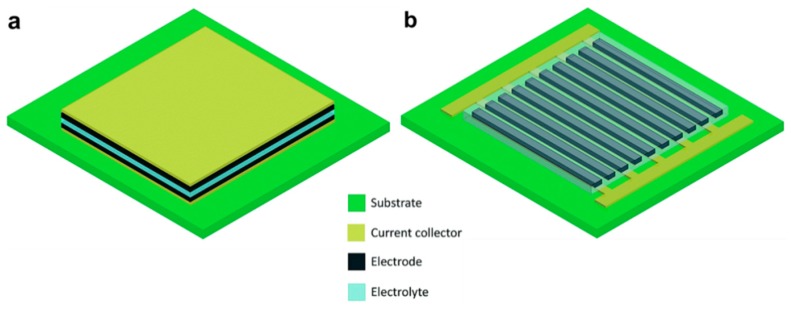
Schematics illustration of (**a**) sandwich and (**b**) planar configurations of micro-supercapacitors. Adapted with permission from Reference [12]. Copyright © 2014 The Royal Society of Chemistry.

**Figure 3 materials-12-00978-f003:**
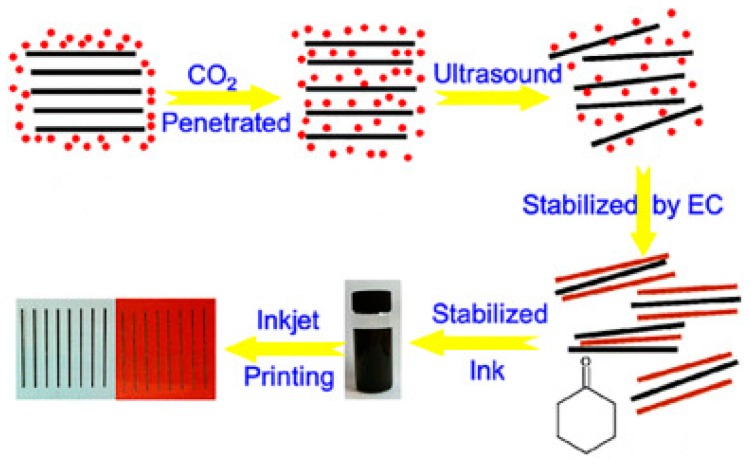
Schematic illustration for the preparation of pristine graphene ink using ultrasound-assisted supercritical CO_2_ and its printed electrodes. Adapted with permission from Reference [82]. Copyright © 2014 American Chemical Society.

**Figure 4 materials-12-00978-f004:**
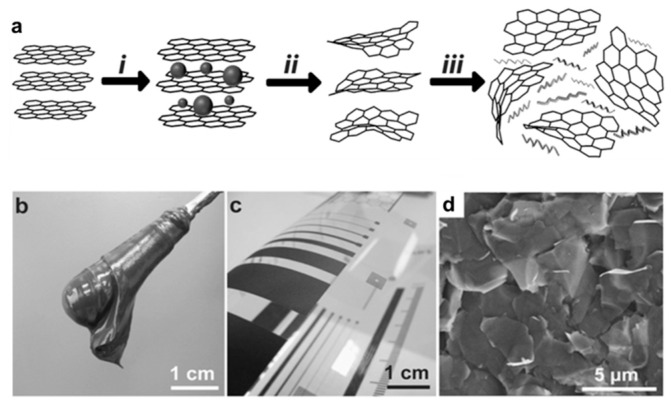
(**a**) Schematic illustration of the (i) graphite intercalation, (ii) thermal expansion, and (iii) graphene gelation; (**b**) image of graphene paste on a spatula; (**c**) poly(ethylene terephthalate) (PET) foil with a test pattern screen printed with graphene paste, (**d**) SEM images of large-area prints at 5000× magnification. Reproduced with permission from Reference [83]. Copyright © 2016 Wiley.

**Figure 5 materials-12-00978-f005:**
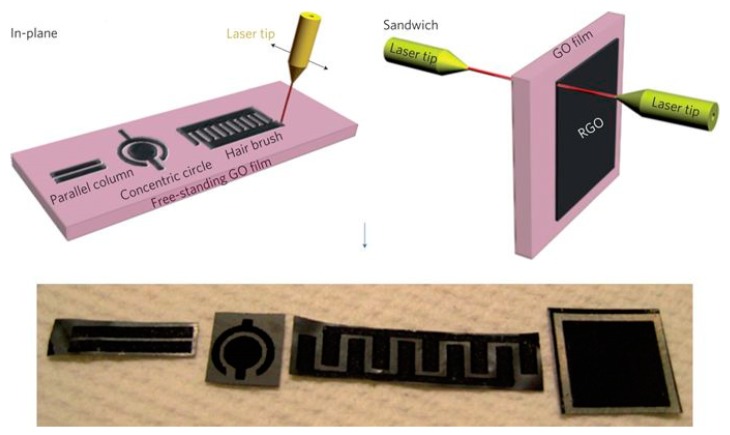
Schematics of laser patterning of free-standing hydrated GO films for fabrication graphene micro-supercapacitor devices with in-plane and sandwich geometries. Reprinted with permission from Reference [106]. Copyright © 2011 Nature Publishing Group.

**Figure 6 materials-12-00978-f006:**
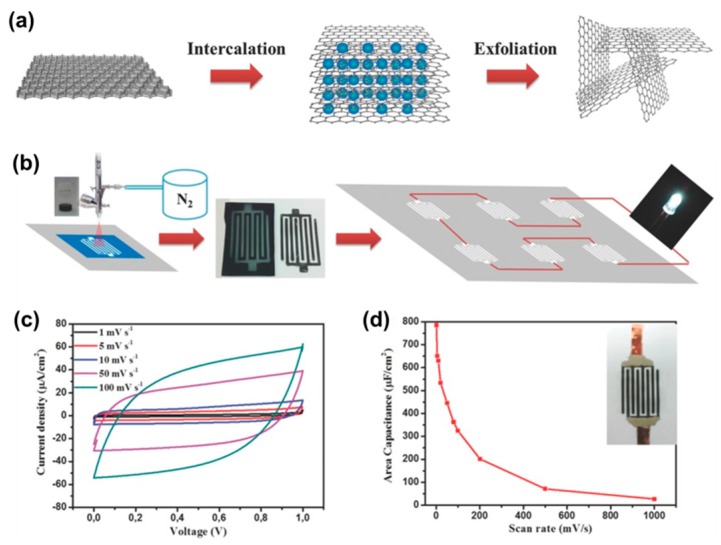
(**a**) Schematic illustration of the electrochemical exfoliation of graphene, (**b**) schematic illustration of the direct printing of single and arrayed micro-supercapacitor devices, (**c**) cyclic voltammetry curves of a printed micro-supercapacitor on a paper substrate, (**d**) the evolution of areal capacitance versus scan rate. Reproduced with permission from Reference [111]. Copyright © 2016 Wiley.

**Figure 7 materials-12-00978-f007:**
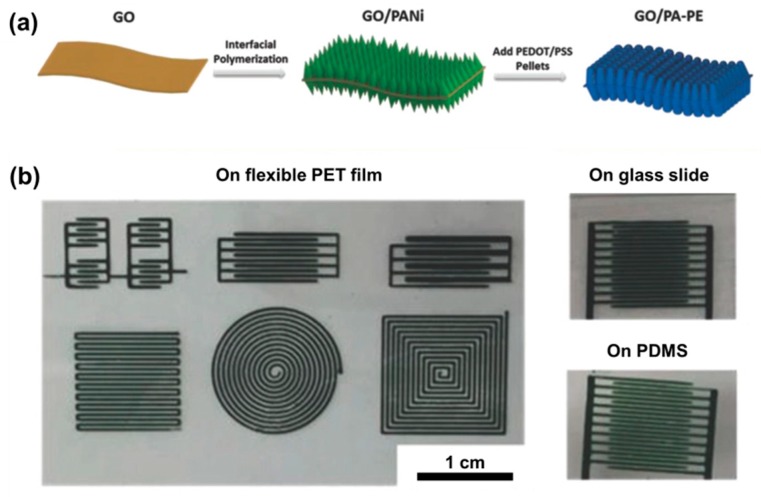
(**a**) Schematic illustration of the preparation of the composite inks. (**b**) Digital photographs of extrusion printed patterns on various substrates. Reproduced with permission from Reference [112]. Copyright © 2018 Wiley.

**Figure 8 materials-12-00978-f008:**
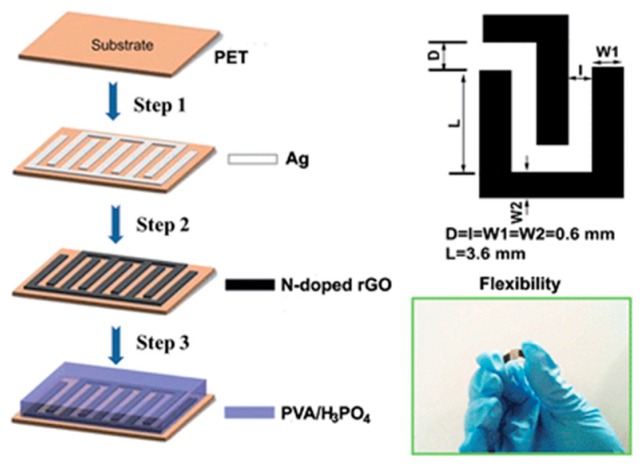
Schematic illustration for the fabrication of flexible solid-state micro-supercapacitor utilizing screen printing. Reproduced with permission from Reference [121]. Copyright © 2014 The Royal Society of Chemistry.

**Figure 9 materials-12-00978-f009:**
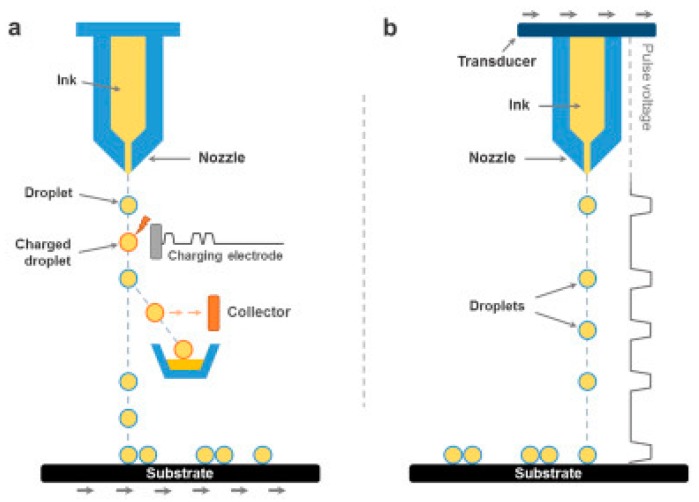
Schematic illustration of the (**a**) continuous mode and (**b**) drops-on-demand mode of inkjet printing. Reproduced with permission from Reference [37]. Copyright © 2018 Elsevier.

**Figure 10 materials-12-00978-f010:**
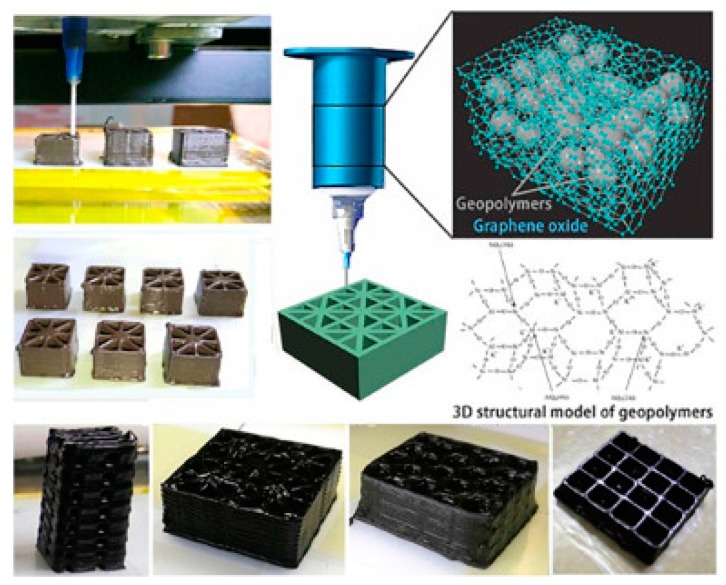
Schematic illustration of the 3D-printing process and some 3D-printed structures. Reprinted with permission from Reference [135]. Copyright © 2017 Elsevier.

**Table 1 materials-12-00978-t001:** Summary of various graphene-based planar micro-supercapacitors derived from graphene oxide.

Reduction Method	Substrate	Electrolyte	Specific Capacitance	Power Density	Reference
Hydrazine at 60 °C	PET film	PVA/H_3_PO_4_	462 μF·cm^−2^	324 W·cm^−3^	[66]
Cu-based redox potential	Polyimide (PI)	PVA/H_2_SO_4_	0.95 mF·cm^−2^		[102]
Hydroiodic acid (HI)	Plastic film	PVA/H_2_SO_4_	41.8 F·cm^−3^	29.2 mW·cm^−2^	[103]
CH_4_ plasma at 700 °C	Silicon wafer	PVA/H_3_PO_4_	80.7 μF·cm^−2^		[104]
Laser writing	Plastic film	PVA/H_2_SO_4_	3.05 F·cm^−3^	30 W·cm^−3^	[105]
Laser writing	GO film	H_2_O	3.1 F·cm^−3^	1.7 W·cm^−3^	[106]

**Table 2 materials-12-00978-t002:** The properties of different printing techniques. Inset: (**a**) Screen printing, reproduced with permission from Reference [115]. Copyright © 2014 The Royal Society of Chemistry. (**b**) Inkjet printing, reproduced with permission from Reference [117]. Copyright © 2018 The Royal Society of Chemistry. (**c**) 3D printing, reproduced with permission from Reference [118]. Copyright © 2016 Wiley.

Printing Techniques	(a) Screen Printing 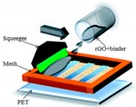	(b) Inkjet Printing 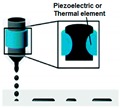	(c) 3D Printing 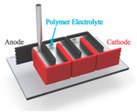
**Ink requirements**	High viscosity Shear thinning	Low viscosity High surface tension	Shear thinning Quick solidification
**Resolution**	~10 µm	~2 µm	~10 µm
**Versatility**	Mask required	Maskless	Maskless
**Printing speed**	Ultrafast	Fast	Slow

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
