# Peer review of "Graphene-Based Inks for Printing of Planar Micro-Supercapacitors: A Review"

_materials, 2019, doi:10.3390/ma12060978_

Round 1
Reviewer 1 Report
Printing techniques have been regarded as a class of promising strategies to fabricate planar energy storage devices, due to cost-effective equipment, simple operation and excellent process flexibility. In the manuscript, the authors tried to review the recent advances of printable planar micro-supercapacitors based on graphene inks. However, the content of the manuscript departed from the theme and there were plenty of detailed mistakes. I recommend the reconsideration of this paper at least after major revisions. Some comments are listed as follows:
1. Title of the manuscript is “graphene-based inks for printing of planar micro-supercapacitors: a review”, in which there are several key words including graphene, inks, printing and planar micro-supercapacitors. Hence, the manuscript should focus on graphene based inks used for printable planar micro-supercapacitors. However, the length of various sections is not appropriate and some sections is irrelevant to the theme. For example,
(1) In section 2, the authors have introduced some basic knowledge including materials, geometry designs and evaluations, which is certainly helpful for readers to understand micro-supercapacitors. However, the length of this part is closed to section 3 or section 4, which confused primary and secondary content of this review. Hence, the authors are suggested to enrich section 3 and section 4 by introducing more typical examples.
(2) In section 3, the authors tried to introduce some graphene based inks for printing. However, there are several problems needing to be pointed out: (i) only inkjet printing is mentioned in the whole section 3, the readers could not achieve information about graphene based inks for other printing strategies; (ii) as pointed by authors, reduction methods has a significant influence on properties of rGO, yet which seems irrelevant to this topic; (iii) similarly, though hybrid composites combining graphene and pseudo-capacitive materials is an effective route to enhance energy density and restacking of graphene sheets is a major reason for performance degeneration of micro-supercapacitors, as emphasized in section 3.3, they are irrelevant to theme of this review. Besides, it is suggested to combine section 3 and section 4 to help readers understand relation between ink properties and printing techniques.
(3) In section 4, various printing techniques have been introduced from aspects of resolution, ink properties and speed. However, as the most important part of this review, the content is very limited and great modification is required. Some suggestions are listed as follows: (i) some typical examples should be introduced in detail in order to help readers understand respect characteristics of different printing techniques, like from line 321 to line 326 in section 4.1. however, in section 4.2 and 4.3, readers could not achieve any information about micro-supercapacitors due to lack of examples; (ii) some latest advances about graphene based inks for printable planar micro-supercapacitors should be involved in this review, such as “ACS Nano 2017, 11, 8249”, “ACS Appl. Mater. Inter. 2017, 9, 37136” and “Energy Environ. Sci. 2019, DOI: 10.1039/C8EE02924E”; (iii) in fact, laser writing should not be involved in printing technologies, especially in ink based printing technologies, so the section 4.4 should be deleted considering the title of this review.
2. There are lots of incorrect or confusing statements, for example,
In line 51, “due to numerous advantages in performance and fabrication”, please give out specific advantages;
In line 80, “reducing the discharge time”, just from formula 3, the power density could be improved by reducing the discharge time. However, discharge time have an influence on E so it is difficult to predict change of power density. Besides, there are multiple factors to evaluate performance of micro-supercapacitors, which restrict each other most of time. Therefore, it is meaningless to just emphasize performance improvement rather than point out specific indicators;
In line 96, “have relatively power”, micro-supercapacitors based EDLC materials should possess higher power and lower energy;
In line 109, “providing much higher energy density in several orders of magnitude”, please give out typical examples;
In line 142, “the interspace between electrodes can be narrowed down to several hundred nanometers”, please check the reference carefully again.
3. There are plenty of detailed mistakes. For example,
In line 24, the abbreviation of “IoT” has been defined, but “Internet of things” is still used in line 25;
In line 63, the “thinner, flexible, rollable, transparent” should be “thin, flexible, rollable, transparent”;
In line 86, the “Base on” should be “Based on”;
In line 87, the “pseodo-capacitive” should be “pseudo-capaciticve”;
In line 88, the “electric double-layer capacitive (EDLC)” is defined, yet it has already appeared in line 87;
In line 96, the “million times” should be “millions of times”;
In line 101, the “provides ions for the charge-storage mechanisms and defining” should be “provides ions for the charge-storage mechanisms and defiens”;
In line 184, the sentence “ecofriendly and have low boiling-point” lack of noun.
Besides above, there are still plenty of unmentioned mistakes. Please carefully check your manuscript before submission.
Author Response
Response to the reviewers’ comments
Reviewer 1:
1. Title of the manuscript is “graphene-based inks for printing of planar micro-supercapacitors: a review”, in which there are several key words including graphene, inks, printing and planar micro-supercapacitors. Hence, the manuscript should focus on graphene based inks used for printable planar micro-supercapacitors. However, the length of various sections is not appropriate and some sections is irrelevant to the theme. For example,
(1) In section 2, the authors have introduced some basic knowledge including materials, geometry designs and evaluations, which is certainly helpful for readers to understand micro-supercapacitors. However, the length of this part is closed to section 3 or section 4, which confused primary and secondary content of this review. Hence, the authors are suggested to enrich section 3 and section 4 by introducing more typical examples.
We thank the reviewer for their comment. We have revised the manuscript accordingly and enriched section 3 & 4 with more examples. The changes are highlighted in blue colour in the revised manuscript.
(2) In section 3, the authors tried to introduce some graphene based inks for printing. However, there are several problems needing to be pointed out: (i) only inkjet printing is mentioned in the whole section 3, the readers could not achieve information about graphene based inks for other printing strategies; (ii) as pointed by authors, reduction methods has a significant influence on properties of rGO, yet which seems irrelevant to this topic; (iii) similarly, though hybrid composites combining graphene and pseudo-capacitive materials is an effective route to enhance energy density and restacking of graphene sheets is a major reason for performance degeneration of micro-supercapacitors, as emphasized in section 3.3, they are irrelevant to theme of this review. Besides, it is suggested to combine section 3 and section 4 to help readers understand relation between ink properties and printing techniques.
(i) We have added more state-of-the-art formulation of graphene inks for screen printing and 3D printing, as recommended by the reviewer.
(ii) We agree that the reduction method could be irrelevant to the printing process. However, reduction is an essential step to restore the electrical properties of graphene and it has a significant influence on the performance of the resulted micro-supercapacitors. We discuss the reduction methods to illustrate the clear picture for the readers to understand the fabrication process of the devices.
(iii) In our opinion, the composite inks of graphene and pseudo-capacitive materials are relevant to the theme of this review. Graphene and other pseudo-capacitive materials can support each other to exploit the full potential of both materials and improve the printability of the inks. For example, instead of using a surfactant, which do not exhibit any interesting electrochemical properties, adding a small amount of PEDT:PSS could not only help to stabilize graphene in the dispersion and tailor it rheology to become a printable ink, but also enhance it electrochemical performance. The graphene composites discuss in this review are printable dispersions of graphene and other pseudo-capacitive materials. We strongly believe that this section could significantly contribute to the knowledge of this field and we project that graphene composite inks could attract enormous interest in the near future.
We thank the reviewer for their suggestion in combining section 3 and 4. As we have enriched these two sections according to the reviewer’s previous comment, and we think that separate section 3 and 4 will help the readers to clearly understand the issues on formulation of graphene inks and the technical challenge depositing them. We also introduced more examples and discussed in detail in both section to help the readers to understand the relationship between the ink properties and the printing techniques.
(3) In section 4, various printing techniques have been introduced from aspects of resolution, ink properties and speed. However, as the most important part of this review, the content is very limited and great modification is required. Some suggestions are listed as follows: (i) some typical examples should be introduced in detail in order to help readers understand respect characteristics of different printing techniques, like from line 321 to line 326 in section 4.1. however, in section 4.2 and 4.3, readers could not achieve any information about micro-supercapacitors due to lack of examples; (ii) some latest advances about graphene based inks for printable planar micro-supercapacitors should be involved in this review, such as “ACS Nano 2017, 11, 8249”, “ACS Appl. Mater. Inter. 2017, 9, 37136” and “Energy Environ. Sci. 2019, DOI: 10.1039/C8EE02924E”; (iii) in fact, laser writing should not be involved in printing technologies, especially in ink based printing technologies, so the section 4.4 should be deleted considering the title of this review.
(i) We have added more typical example and discussed in detail characteristics of different printing techniques in section 4. We also enriched the knowledge of graphene ink formulation in section 3 by adding more typical examples. The changes are highlighted in blue colour in the revised manuscript.
(ii) We have added more references and discussed in detail state-of-the-art formulation of graphene inks and their printing techniques for fabrication of planar micro-supercapacitors including the suggested papers from the reviewer.
(iii) We agree that laser writing should not be involved in printing technologies. We have deleted section 4.4 (laser writing) according to the reviewer comment.
2. There are lots of incorrect or confusing statements, for example,
In line 51, “due to numerous advantages in performance and fabrication”, please give out specific advantages;
Benefiting from the narrow gap between the interdigital electrodes and the simple design, micro-supercapacitors with in-plane interdigital design possess several advantages compared to the conventional sandwich structure includes: ease of fabrication, better flexibility, higher mechanical performance, more stable under various operating conditions, higher rate performance and frequency response. The statement in line 51 is a short introduction to the micro-supercapacitors with planar design. Its specific advantages have been discussed in detail in section 2.2 (designs of micro-supercapacitors).
In line 80, “reducing the discharge time”, just from formula 3, the power density could be improved by reducing the discharge time. However, discharge time have an influence on E so it is difficult to predict change of power density. Besides, there are multiple factors to evaluate performance of micro-supercapacitors, which restrict each other most of time. Therefore, it is meaningless to just emphasize performance improvement rather than point out specific indicators;
We agree with the reviewer that there are multiple factors to evaluate performance of micro-supercapacitors, and they restricted each other most of the time. We are not stating that reducing the discharge time will result in exponential increase in energy density, but reducing the discharge time does enhance the electrochemical performance of the device, for example, higher rate performance for ultrafast micro-supercapacitors. A lot of interest has also been devoted onto this regards. We mentioned the discharge time as it provided fundamental guidance for choosing materials and designs for micro-supercapacitors. We have revised the context in the revised manuscript to ensure that it’s not confusing to the readers.
In line 96, “have relatively power”, micro-supercapacitors based EDLC materials should possess higher power and lower energy;
We have revised the manuscript according to the reviewer’s comment.
In line 109, “providing much higher energy density in several orders of magnitude”, please give out typical examples;
We have revised the manuscript and provided some typical references.
In line 142, “the interspace between electrodes can be narrowed down to several hundred nanometers”, please check the reference carefully again.
In 2010, Zhang and co-workers (Nano Today 5.1 (2010): 15-20) has successfully imprinted graphene microcircuits onto graphene oxide films via direct femtosecond laser reduction process, affording high-resolution circuits with the width of about 500 nm. With the scanning step length can be programed down to 100 nm, this work showed the possibility to narrow down the interspace the interspace between electrodes to several hundred nanometers. This reference has been added into the above mentioned statement in the revised manuscript.
3. There are plenty of detailed mistakes. For example,
In line 24, the abbreviation of “IoT” has been defined, but “Internet of things” is still used in line 25;
In line 63, the “thinner, flexible, rollable, transparent” should be “thin, flexible, rollable, transparent”;
In line 86, the “Base on” should be “Based on”;
In line 87, the “pseodo-capacitive” should be “pseudo-capaciticve”;
In line 88, the “electric double-layer capacitive (EDLC)” is defined, yet it has already appeared in line 87;
In line 96, the “million times” should be “millions of times”;
In line 101, the “provides ions for the charge-storage mechanisms and defining” should be “provides ions for the charge-storage mechanisms and defiens”;
In line 184, the sentence “ecofriendly and have low boiling-point” lack of noun.
Besides above, there are still plenty of unmentioned mistakes. Please carefully check your manuscript before submission.
We have extensively revised the manuscript and corrected all the grammatical mistakes including the above mentioned mistakes pointed out by the reviewer.
Reviewer 2:
The authors reported a review concerning the potential of graphene-based inks for micro-supercapacitor applications. The study is categorized into different sections as follows. First, an introduction about the importance and potential of micro-supercapacitors in a wide range of technological applications, as for example internet of things (IoT), is presented. Next, an explanation about the characteristics of a micro-supercapacitor device and its evaluation in terms of electrochemical performance is discussed. Lately, a summary of several types of graphene-based inks and the different printing techniques to deposit them is reported according to the state-of-the-art. Finally, a brief discussion about their electrochemical performances and their potential in the domain of micro-supercapacitor is analyzed. In my opinion, the work is well structured and organized, and the reading was clear and fluent. I appreciated enormously this aspect. In addition, the research topic presented in this study is very exciting with a great potential in the field of electrochemical energy storage devices. However, I consider this work presents major concerns regarding the content, the bibliography research and the evaluation of the electrochemical performances, which are detailed below:
General comments:
Some misprints were detected throughout the text as for example pag. 3 base on or pseodo-cap. I have cited only some examples. Please revise the manuscript carefully.
We have extensively revised the manuscript and corrected all the misprints including the above mentioned example pointed out by the reviewer. The changes are highlighted in blue colour in the revised manuscript.
The references should be adapted according to the journal’s format. Please revise this section carefully.
We have reformatted the references and their EndNote style according to the journal’s format.
Technical comments:
I agree with the authors this study is focused mainly on the potential of graphene-based inks for micro-supercapacitor applications, but the authors reported a paragraph about the potential of graphene in the introduction section, in a general way, in this field. I think the authors should highlight at least one of the most promising graphene nanostructures for micro-supercapacitors, vertically-oriented graphene nanosheets. This structure deserves a special attention and probably a specific mention is recommended due to its enormous interest in recent years. In this regard, the introduction could be defined differently with a special focus on the different graphene (nano)-structures in the domain of micro-supercapacitors.
We thank the reviewer for their comment. We have revised the introduction part according the reviewer’s comment with a special focus on different graphene nanostructures and specific emphasize on vertically-oriented graphene.
Regarding the printing techniques for graphene structures and derivatives (Section 4), new concepts dealing with the extrusion printing technique have been analyzed, especially for 3D printing. In my opinion, a particular study about this technique should be discussed in detail. Just some references are cited as an example, Adv. Funct. Mater. 2018, 28, 1706592. In this direction, 3D printing (sub-section 4.3.3), some concepts regarding also fused depositing modeling (FDM) should be investigated among others (Synth. Metals. 2016, 217, 79). Within this context, a special focus is devoted to the state-of-the-art, which I consider one of the most important weakness of this study. Thus, important references are missing. More specifically, the section printing techniques should be analyzed in-depth according to the literature since an update bibliography research is highly recommended.
Ultrahigh-voltage integrated micro-supercapacitors with designable shapes and superior flexibility, Energy Environmental Science 2019
Scalable Fabrication and Integration of Graphene Micro-Supercapacitors through Full Inkjet Printing, ACS Nano 2017, 11, 8249
Ultraflexible In‐Plane Micro‐Supercapacitors by Direct Printing of Solution‐Processable Electrochemically Exfoliated Graphene Adv. Mater. 2016, 28, 2217-2222
…. (Just some examples, revise the literature carefully)
Please, revise the literature of the section 4 (Printing techniques) and correlate the technique with the corresponding electrochemical performances. A detailed electrochemical characterization should be provided to estimate which is/are the best method(s).
We have added more references and discussed in detail the state-of-the-art printing techniques for graphene inks including the suggested papers from the reviewer.
Finally, an analysis of the electrochemical performances concerning the different graphene structures presented in this study is necessary. Only a table was reported describing just specific capacitance and power density (Table 2). Lately, a brief discussion about some electrochemical performances from the printing techniques was discussed. In my opinion, a completed and detailed analysis is highly recommended to evaluate the potential of ink-based graphene for micro-supercapacitor applications. Consequently, I propose a summary state-of-the-art table and a Ragone plot to highlight the reason of using graphene-based inks and its corresponding deposition method. A better, clear and detailed electrochemical performances in terms of E, P, C and lifetime is necessary.
We agree that electrochemical performance (E, P, C and cycle life) are among the most important facets when study the supercapacitor. However, as the research in this field still in its early stage, research on graphene inks are still mainly focused on formulation of stable graphene dispersions with printability and demonstration of simple conductive circuits, with less attention has been paid to their electrochemical performance.
There is still a scarce of literature on graphene inks for printing of planar micro-supercapacitors. We have gone through the literature carefully and realized that many graphene inks for supercapacitors reported to date were realized in sandwich design, which is not relevant to the theme of this review. Even in many reports where graphene inks was used to print “planar” micro-supercapacitors, the performance metrics were not uniformly reported. Some references reported gravimetric and volumetric performances, only a limited number of references reported the areal metric performance of the printed devices, not enough information to make a Ragone plot or a comparable table.
To this end, we have discussed in detail the performance of some typical graphene inks formulation as well as printing techniques in section 3 and 4, provided a clear picture about the performance of state-of-the-art devices. We also review a proper way for reporting the performance of planar micro-supercapacitors based on areal metric of the devices. This is very useful and it will guide future research in this field to uniformly report the performance of planar micro-supercapacitors.
In conclusion, this work requires an exhaustive and detailed state-of-the-art according to the research topic presented. Furthermore, a better description concerning the electrochemical performances of the different graphene structures is highly recommended. This particular point has not been well analyzed. In this direction, important efforts should be still addressed to improve the content of this study according to the quality of the journal of Materials.
We thank the reviewer for their comment.

Reviewer 2 Report
The authors reported a review concerning the potential of graphene-based inks for micro-supercapacitor applications. The study is categorized into different sections as follows. First, an introduction about the importance and potential of micro-supercapacitors in a wide range of technological applications, as for example internet of things (IoT), is presented. Next, an explanation about the characteristics of a micro-supercapacitor device and its evaluation in terms of electrochemical performance is discussed. Lately, a summary of several types of graphene-based inks and the different printing techniques to deposit them is reported according to the state-of-the-art. Finally, a brief discussion about their electrochemical performances and their potential in the domain of micro-supercapacitor is analyzed. In my opinion, the work is well structured and organized, and the reading was clear and fluent. I appreciated enormously this aspect. In addition, the research topic presented in this study is very exciting with a great potential in the field of electrochemical energy storage devices. However, I consider this work presents major concerns regarding the content, the bibliography research and the evaluation of the electrochemical performances, which are detailed below:
General comments:
Some misprints were detected throughout the text as for example pag. 3 base on or pseodo-cap. I have cited only some examples. Please revise the manuscript carefully.
The references should be adapted according to the journal’s format. Please revise this section carefully.
Technical comments:
I agree with the authors this study is focused mainly on the potential of graphene-based inks for micro-supercapacitor applications, but the authors reported a paragraph about the potential of graphene in the introduction section, in a general way, in this field. I think the authors should highlight at least one of the most promising graphene nanostructures for micro-supercapacitors, vertically-oriented graphene nanosheets. This structure deserves a special attention and probably a specific mention is recommended due to its enormous interest in recent years. In this regard, the introduction could be defined differently with a special focus on the different graphene (nano)-structures in the domain of micro-supercapacitors.
Regarding the printing techniques for graphene structures and derivatives (Section 4), new concepts dealing with the extrusion printing technique have been analyzed, especially for 3D printing. In my opinion, a particular study about this technique should be discussed in detail. Just some references are cited as an example, Adv. Funct. Mater. 2018, 28, 1706592. In this direction, 3D printing (sub-section 4.3.3), some concepts regarding also fused depositing modeling (FDM) should be investigated among others (Synth. Metals. 2016, 217, 79). Within this context, a special focus is devoted to the state-of-the-art, which I consider one of the most important weakness of this study. Thus, important references are missing. More specifically, the section printing techniques should be analyzed in-depth according to the literature since an update bibliography research is highly recommended.
Ultrahigh-voltage integrated micro-supercapacitors with designable shapes and superior flexibility, Energy Environmental Science 2019
Scalable Fabrication and Integration of Graphene Micro-Supercapacitors through Full Inkjet Printing, ACS Nano 2017, 11, 8249
Ultraflexible In‐Plane Micro‐Supercapacitors by Direct Printing of Solution‐Processable Electrochemically Exfoliated Graphene Adv. Mater. 2016, 28, 2217-2222
…. (Just some examples, revise the literature carefully)
Please, revise the literature of the section 4 (Printing techniques) and correlate the technique with the corresponding electrochemical performances. A detailed electrochemical characterization should be provided to estimate which is/are the best method(s).
Finally, an analysis of the electrochemical performances concerning the different graphene structures presented in this study is necessary. Only a table was reported describing just specific capacitance and power density (Table 2). Lately, a brief discussion about some electrochemical performances from the printing techniques was discussed. In my opinion, a completed and detailed analysis is highly recommended to evaluate the potential of ink-based graphene for micro-supercapacitor applications. Consequently, I propose a summary state-of-the-art table and a Ragone plot to highlight the reason of using graphene-based inks and its corresponding deposition method. A better, clear and detailed electrochemical performances in terms of E, P, C and lifetime is necessary.
In conclusion, this work requires an exhaustive and detailed state-of-the-art according to the research topic presented. Furthermore, a better description concerning the electrochemical performances of the different graphene structures is highly recommended. This particular point has not been well analyzed. In this direction, important efforts should be still addressed to improve the content of this study according to the quality of the journal of Materials.
Author Response
Reviewer 2:
The authors reported a review concerning the potential of graphene-based inks for micro-supercapacitor applications. The study is categorized into different sections as follows. First, an introduction about the importance and potential of micro-supercapacitors in a wide range of technological applications, as for example internet of things (IoT), is presented. Next, an explanation about the characteristics of a micro-supercapacitor device and its evaluation in terms of electrochemical performance is discussed. Lately, a summary of several types of graphene-based inks and the different printing techniques to deposit them is reported according to the state-of-the-art. Finally, a brief discussion about their electrochemical performances and their potential in the domain of micro-supercapacitor is analyzed. In my opinion, the work is well structured and organized, and the reading was clear and fluent. I appreciated enormously this aspect. In addition, the research topic presented in this study is very exciting with a great potential in the field of electrochemical energy storage devices. However, I consider this work presents major concerns regarding the content, the bibliography research and the evaluation of the electrochemical performances, which are detailed below:
General comments:
Some misprints were detected throughout the text as for example pag. 3 base on or pseodo-cap. I have cited only some examples. Please revise the manuscript carefully.
We have extensively revised the manuscript and corrected all the misprints including the above mentioned example pointed out by the reviewer. The changes are highlighted in blue colour in the revised manuscript.
The references should be adapted according to the journal’s format. Please revise this section carefully.
We have reformatted the references and their EndNote style according to the journal’s format.
Technical comments:
I agree with the authors this study is focused mainly on the potential of graphene-based inks for micro-supercapacitor applications, but the authors reported a paragraph about the potential of graphene in the introduction section, in a general way, in this field. I think the authors should highlight at least one of the most promising graphene nanostructures for micro-supercapacitors, vertically-oriented graphene nanosheets. This structure deserves a special attention and probably a specific mention is recommended due to its enormous interest in recent years. In this regard, the introduction could be defined differently with a special focus on the different graphene (nano)-structures in the domain of micro-supercapacitors.
We thank the reviewer for their comment. We have revised the introduction part according the reviewer’s comment with a special focus on different graphene nanostructures and specific emphasize on vertically-oriented graphene.
Regarding the printing techniques for graphene structures and derivatives (Section 4), new concepts dealing with the extrusion printing technique have been analyzed, especially for 3D printing. In my opinion, a particular study about this technique should be discussed in detail. Just some references are cited as an example, Adv. Funct. Mater. 2018, 28, 1706592. In this direction, 3D printing (sub-section 4.3.3), some concepts regarding also fused depositing modeling (FDM) should be investigated among others (Synth. Metals. 2016, 217, 79). Within this context, a special focus is devoted to the state-of-the-art, which I consider one of the most important weakness of this study. Thus, important references are missing. More specifically, the section printing techniques should be analyzed in-depth according to the literature since an update bibliography research is highly recommended.
Ultrahigh-voltage integrated micro-supercapacitors with designable shapes and superior flexibility, Energy Environmental Science 2019
Scalable Fabrication and Integration of Graphene Micro-Supercapacitors through Full Inkjet Printing, ACS Nano 2017, 11, 8249
Ultraflexible In‐Plane Micro‐Supercapacitors by Direct Printing of Solution‐Processable Electrochemically Exfoliated Graphene Adv. Mater. 2016, 28, 2217-2222
…. (Just some examples, revise the literature carefully)
Please, revise the literature of the section 4 (Printing techniques) and correlate the technique with the corresponding electrochemical performances. A detailed electrochemical characterization should be provided to estimate which is/are the best method(s).
We have added more references and discussed in detail the state-of-the-art printing techniques for graphene inks including the suggested papers from the reviewer.
Finally, an analysis of the electrochemical performances concerning the different graphene structures presented in this study is necessary. Only a table was reported describing just specific capacitance and power density (Table 2). Lately, a brief discussion about some electrochemical performances from the printing techniques was discussed. In my opinion, a completed and detailed analysis is highly recommended to evaluate the potential of ink-based graphene for micro-supercapacitor applications. Consequently, I propose a summary state-of-the-art table and a Ragone plot to highlight the reason of using graphene-based inks and its corresponding deposition method. A better, clear and detailed electrochemical performances in terms of E, P, C and lifetime is necessary.
We agree that electrochemical performance (E, P, C and cycle life) are among the most important facets when study the supercapacitor. However, as the research in this field still in its early stage, research on graphene inks are still mainly focused on formulation of stable graphene dispersions with printability and demonstration of simple conductive circuits, with less attention has been paid to their electrochemical performance.
There is still a scarce of literature on graphene inks for printing of planar micro-supercapacitors. We have gone through the literature carefully and realized that many graphene inks for supercapacitors reported to date were realized in sandwich design, which is not relevant to the theme of this review. Even in many reports where graphene inks was used to print “planar” micro-supercapacitors, the performance metrics were not uniformly reported. Some references reported gravimetric and volumetric performances, only a limited number of references reported the areal metric performance of the printed devices, not enough information to make a Ragone plot or a comparable table.
To this end, we have discussed in detail the performance of some typical graphene inks formulation as well as printing techniques in section 3 and 4, provided a clear picture about the performance of state-of-the-art devices. We also review a proper way for reporting the performance of planar micro-supercapacitors based on areal metric of the devices. This is very useful and it will guide future research in this field to uniformly report the performance of planar micro-supercapacitors.
In conclusion, this work requires an exhaustive and detailed state-of-the-art according to the research topic presented. Furthermore, a better description concerning the electrochemical performances of the different graphene structures is highly recommended. This particular point has not been well analyzed. In this direction, important efforts should be still addressed to improve the content of this study according to the quality of the journal of Materials.
We thank the reviewer for their comment.

Round 2
Reviewer 2 Report
The authors revised the manuscript according to my comments carefully. Consequently, I accept this manuscript for publication in the journal Materials